# Sinonasal Meningioma in a Siberian Tiger (*Panthera tigris altaica*)

**DOI:** 10.3390/vetsci9090457

**Published:** 2022-08-25

**Authors:** Louise van der Weyden, Peter Caldwell, Christine Steyrer, Nicolize O’Dell, Alischa Henning

**Affiliations:** 1Wellcome Sanger Institute, Wellcome Genome Campus, Cambridge CB10 1SA, UK; 2Old Chapel Veterinary Clinic, Villieria, Pretoria 0186, South Africa; 3Lionsrock Big Cat Sanctuary, Bethlehem 9700, South Africa; 4Department of Paraclinical Sciences, Faculty of Veterinary Science, University of Pretoria, Onderstepoort 0110, South Africa; 5Centre for Veterinary Wildlife Studies, Faculty of Veterinary Sciences, University of Pretoria, Onderstepoort 0110, South Africa

**Keywords:** tiger, tumour, meningioma, transitional, sinonasal

## Abstract

**Simple Summary:**

Meningiomas are the most common primary brain tumour in dogs and cats. However, whilst there are numerous reports of meningiomas at extracranial sites in the dog (such as the spine, the eye and near the nasal cavity), in cats there have only been a few case reports of meningiomas arising in the spine, and no reports of post-mortem confirmed meningiomas arising in the eyes or near the nasal cavity. In this report, a 20-year-old captive tiger (*Panthera tigris altaica*) with a history of chronic eye inflammation, resulting in eventual removal of the eye, spontaneously developed epilepsy. Over the course of 2 years, the seizures worsened to the point where the animal was eventually euthanized. At autopsy, a mass was found near the nasal cavity and histological analysis showed tumour cells surrounded by a collagenous matrix. The diagnosis was sinonasal transitional meningioma. This is the first report of a captive wild felid with an extracranial meningioma, specifically a tiger with a sinonasal transitional meningioma.

**Abstract:**

Meningiomas are the most common primary brain tumour in dogs and cats. However, whilst there are numerous reports of extracranial (spinal, orbital and sinonasal) meningiomas in the dog, there have only been a few case reports of spinal meningiomas, and no post-mortem confirmed orbital or sinonasal meningiomas in cats. In this report, a 20-year-old captive tiger (*Panthera tigris altaica*) with a history of chronic ocular inflammation resulting in enucleation, spontaneously developed tetanic convulsions (epileptic seizures) that over a 2-year period resulted in a gradually worsening condition and the animal was eventually euthanized. At autopsy, a focal, expansile, neoplastic mass was found in the caudal nasal cavity midline, abutting the cribriform plate and slightly compressing the calvarium. Histological analysis revealed nasal turbinates attached to a well-circumscribed expansile multi-lobular mass consisting of interlacing whorls and streams of neoplastic cells supported by a variably fibrous to microcystic collagenous matrix displaying rare psammoma bodies. The diagnosis was sinonasal transitional meningioma. This is the first report of a captive wild felid with an extracranial meningioma, specifically a tiger with a sinonasal transitional meningioma.

## 1. Introduction

Meningiomas are tumours that arise from the meninges, which are composed of three layers (dura mater, arachnoid mater and pia mater), and are the most common tumour of the central nervous system in humans. Meningiomas are the most common primary brain tumour and are only rarely found extracranially; <2 % of all meningiomas arise extracranially [1] with the most frequent location being the orbital cavity [2]. There have been a few case reports of nasal meningioma and meningioma of the sinonasal sinuses in humans [3,4,5]. Meningiomas are the most common primary intracranial tumour in dogs and cats [6,7,8,9], with spinal meningiomas occurring less frequently [10,11]. There are case reviews of orbital meningioma (*n* = 22) [12] and sinonasal meningiomas (*n* = 10) [13] in dogs, however, there have been no reports in cats to-date. There has been one report of a meningioma with extensive nasal involvement in a cat; a transitional (mixed) meningioma in a 9-year-old male neutered domestic longhair cat [14]. However, it is important to note that in this case a post-mortem examination was not performed and the authors state it was not possible to rule out that the tumour could represent nasal invasion from an intracranial meningioma or a sinonasal meningioma expanding into the calvarium.

In contrast to domestic cats, there have only been 2 reports of meningiomas in captive wild felids to-date. One was an 18-year old female cougar (*Puma concolor*) from a German zoo that was euthanised and at autopsy was found to have a psammomatous meningioma along with thyroid gland adenocarcinoma, haemangiosarcoma and T-cell lymphoma (however, as this was part of a retrospective study looking at pathology of wild felids at German zoological gardens between 2004–2013, no clinical or histopathological details were provided) [15]. The other was a case report of a 17-year-old female ovariectomized Bengal tiger (*Panthera tigris*
*tigris*) from a big cat sanctuary in the USA that suffered two seizures, expiring shortly after the second one, and at autopsy was found to have a meningioma attached to the left frontal bone [16]. We present the first report of a captive wild felid with an extracranial meningioma, specifically a tiger with a sinonasal transitional meningioma

## 2. Case Presentation

A 15-year-old female intact Siberian tiger was rescued from a privately run big cat facility in Nijeberkoop in the Netherlands, where she was used for breeding purposes, and brought to South Africa to live at Lionsrock Big Cat Sanctuary. Later that year the tiger developed a deep corneal ulcer on her left eye that was treated with conjunctival flap surgery, however, the condition did not resolve and enucleation was performed one month later. Histopathological examination of the eye ball found chronic severe pyogranulomatous conjunctivitis/blepharitis with granulation tissue formation and chronic corneal fibrosis, but no signs of neoplasia in the sections. Nine months later the tiger developed a deep corneal ulcer of the right eye and conjunctival flap surgery was performed. Fourteen months later a new corneal ulcer developed, lateral to the previous surgery (towards the lateral canthus), and a second conjunctival flap was performed. In addition, the tiger was placed on anti-histamine therapy to prevent continual corneal ulcerations.

Three months later, the tiger started displaying twitching of the face and forelimbs with intentional tremors (muscle fasciculations) in the face and neck, bilateral forelimb ataxia, disorientation, and it was also noted that the skin on the medial canthus of the eye appeared to bulge (to at least the size of a hazel nut). In addition, generalized tonic clonic seizures also occurred; sometimes triggered by loud noises or frightening events. Observations carried out over the next two years, both by staff and movement-triggered camera traps, revealed a total of 11 seizures with convulsions, sometimes lasting up to 2 min, in addition to other neurological signs such as facia twitching or tremors. Over the course of the two years, the following medication was administered: Neurontin (600 mg/tab Gabapentin, Pfizer; starting on 5 mg/kg twice a day (bid) for 5 days, then 10 mg/kg bid, then tapering to once a day (oid) after two weeks, maintained at this dose for 4 months), Medrol 104 (16 mg/tablet methylprednisolone; 1 mg/kg bid for 5 days, then 1 mg/kg oid for 14 days, 0.5 mg/kg oid for 14 days, 0.5 mg/kg every other day (eod) for 1 month), Lyrica (75 mg/tab Pregabalin; 1 mg/kg bid for 3 weeks), Purata (30 mg/tab Oxazepam, Aspen; 0.2 mg/kg bid for 10 days, then 0.2 mg/kg oid for 10 days, then 0.2 mg/kg three times a week), Phenobarbitone (30 mg/tablet BeTabs Sedabarb; 3 mg/kg for 1 year), Clavumox (Amoxicillin/clavulanic acid 1 g/tablet; 12.5 mg/kg bid for 14 days), Prednisolone (5 mg/tablet, Aspen; 1 mg/kg bid for 5 days, then 1 mg/kg oid for 14 days, 0.5 mg/kg oid for 14 days, 0.5 mg/kg eod for 1 month), Omeprazole (40 mg/tablet compounded vetscripts tablets; 0.5 mg/kg eod for 6 months), Keppra (750 mg/tablet Levetiracetam, Glaxo Smith Kline; 20 mg/kg bid for one year), Karsivan 106 (50 mg Propentofylline/tablet, MSD animal health; 3 tablets bid) and various supplements to support the liver, including SAMe Solal (s-adenosyl-l-methionine disulfate p-toluensulfonate) and Liver Complex (Ingredients per capsule: Vitamin B1 (Thiamine), Vitamin B2 (Riboflavin), Vitamin B3 (Nicotinamide), Vitamin B5 (Pantothenic Acid), Vitamin B6 (Pyridoxine), Vitamin B12 (Cyanocobalamin), Silybum Marianum (Milk Thistle) Herb Powder, Lecithin (Phospholipids), Silybum Marianum (Milk Thistle) Seed Extract, Cynara Scolymus (Artichoke) Leaf Powder Extract 4:1, Schisandra spp. Fruit Powder, Hordeum Vulgare (Barley) Leaf Powder (SOD Precursor), Taraxacum Officinale (Dandelion) Root Powder, L-Glutathione, Selenium and Gingko biloba).

During this period the tiger was immobilised for evaluation by a veterinarian on three separate occasions, however, on each occasion no abnormalities were detected in the chest x-rays or pupillary light reflex tests. Blood was also taken on each occasion and analysed using a Catalyst One Chemistry Analyzer and ProCyte Dx™ (IDEXX Laboratories) for clinical chemistry profiles (including ALB, ALN/GLOB, ALKP, ALT, AMYL, BUN, BUN/CREA, Ca, CHOL, CREA, GGT, GLOB, GLU, LIPA, PHOS, TBIL and TP), electrolyte profiles (including Cl, K, Na and Na), haematology profiles (full blood count) and levels of SDMA (as a marker of kidney function), BNP (as a marker of heart failure), fasting serum bile acids, serum Amyloid A (as a marker of inflammation), total T4, total TSH, vitamin A and serum phenobarbitone; no abnormalities were detected other than elevated creatinine levels on the third occasion. Due to a brain tumour being suspected, the tiger was treated palliatively and symptomatically with oral corticosteroids and antiepileptic drugs to reduce the rate of growth of the tumour and to keep the siezures under control. This treatment protocol seemed to help with varying levels of success over time as the dosages were adjusted according to the tiger’s symptoms. However, the tiger eventually became refractory to all medications and the seizures became uncontrollable. Thus after several days of inappetence, lethargy, difficulty in administering medication and unresponsiveness to the medication, the decision was taken to euthanise the tiger.

At autopsy, a general examination found the tiger to be in very good body condition, with very large visceral and subcutaneous fat stores, and well-developed skeletal muscle mass. Marked acute diffuse tissue congestion with pulmonary and cerebral oedema (due to euthanasia) and advanced autolysis and putrefaction (due to post-mortem changes) were noted. Upon opening the skull along the midline, a focal, expansile, firm, pale tan, multinodular, cauliflower-like mass (4 × 3 cm) was found in the caudal nasal cavity midline, abutting the cribriform plate (Figure 1a). After partially dissecting and reflecting the mass rostrally, slight compression of the calvarium by the neoplastic mass could be appreciated (Figure 1b). The mass, together with the brain, spinal cord, thyroid, parathyroid, heart, lungs, liver, kidneys, spleen, pancreas, stomach and uterus were all sampled for histopathological examination in 10% buffered formalin. The formalin-fixed tissues were embedded in paraffin wax, sectioned and stained with haematoxylin and eosin. The urine was yellow and cloudy with floccules and no parasitic ova were detected on a faecal flotation. A sample of peripheral blood was taken to generate a smear, which showed no obvious abnormalities.

Histopathological analysis of the mass showed nasal turbinates attached to a well-circumscribed expansile multi-lobular mass consisting of tight to loosely packed interlacing streams and whorls of neoplastic cells supported by a variably fibrous to microcystic collagenous matrix (Figure 2a). The neoplastic cells had indistinct margins, small amounts of pale basophilic cytoplasm, large oval hypochromatic nuclei with finely stippled to vesicular chromatin and few small basophilic nucleoli (Figure 2b). There were rare psammoma bodies noted (Figure 2c) and a few scattered foci of spontaneous necrosis associated with mild haemorrhage and neutrophil infiltration (Figure 2d). The mitotic count averaged one mitosis per 10 high power fields. A diagnosis of transitional meningioma was made and this could be further classified as sinonasal due to the absence of brain involvement.

The brain showed marked neuronal atrophy and mild gliosis, as well as mild multifocal meningeal melanosis. Both kidneys showed pale tan radial medullary streaks of moderate fibrosis (confirmed by Masson’ trichrome stain) with mildly thickened Bowman’s capsules, a few obsolescent glomeruli, mild tubular atrophy (confirmed by periodic acid Schiff stain) and moderate subacute lymphoplasmacytic interstitial cortical nephritis (especially at the corticomedullary junction). There were also scattered renal cortical tubules containing amorphous, pale yellow-pigmented material (likely bile) and a few mineralised foci in renal tubular lumina. The pancreas showed moderate multifocal subacute lymphoplasmacytic inflammation with associated mild pancreatic fat necrosis as well as moderate nodular hyperplasia. The spleen showed moderate lymphoid atrophy with mild haemosiderosis. The lungs showed mildly hyperplastic terminal bronchiolar smooth muscle and moderate anthrasilicosis. The uterus showed marked papillary hyperplasia often with smooth muscle cores (polyps) with mildly hyperplastic and cystic glands. The parathyroid gland showed a prominent C-cell complex. Apart from post-mortem change and congestion, sections of the heart and thyroid gland were unremarkable and sections of liver and gastric mucosa were largely too autolysed to evaluate (although a soft yellow lipoma (1 × 2 cm) was noted on the edge of one liver lobe). All of the tissues that were examinable, did not show the presence of any neoplastic cells, apart from the nasal cavity mass, suggesting that metastasis of the sinonasal meningioma had not occurred, consistent with the typically low grade and expansive, rather than infiltrative growth nature, of most meningiomas in dogs and cats [reviewed in 9].

## 3. Discussion

Meningioma is the most common primary brain tumour in domestic cats and is typically slow growing slow growing with rare infiltrative growth and invasion into the underlying brain parenchyma [8]. Nevertheless, although infiltrative meningiomas are rarely seen in cats, they can co-occur with non-infiltrative meningiomas [17]. Moreover, multiple meningiomas are common in cats (2–5 tumours per cat) [8,18], whereas it is uncommon for the presence of more than one meningioma in dogs [reviewed in 9]. It is also important to note that meningiomas can occur extra-cranially, however, whereas spinal meningiomas, orbital meningiomas and sinonasal meningiomas have been reported in dogs, spinal meningiomas are rare in cats and there are as yet no published reports of orbital and sinonasal meningiomas in cats [reviewed in 9]. Meningioma in domestic cats typically occurs in those over 10 years of age [8,19]. Occurrence in older cats would also seem to be the case for captive wild felids, with the tiger in this case report being 20 years of age, and the only other 2 reports of meningioma in captive wild felids being an 18-year old cougar [15] and a 17-year old Bengal tiger [16].

Clinical signs of meningiomas in domestic cats are highly variable, depending on their location. Single or multiple intracranial tumours can either cause neurological symptoms or be incidental findings [9], with up to 20% of feline meningiomas possibly being clinically silent [20]. Symptoms, when present, loss of consciousness, ataxia and behavioural changes are common [8], however, whereas seizures are the most commonly seen sign in dogs, they are not as commonly seen in cats (15–22% of cases) [8,19]. Interestingly, a positive association has been reported in cats between the occurrence of seizures and the presence of a tumour in the forebrain [19]. Consistent with this, a Bengal tiger with a meningioma in the left frontal lobe had two severe tonic-clonic seizures on the morning of expiration; the first seizure resolved but a few minutes later a second seizure occurred and the tiger expired, with autopsy revealing the presence of a 3 × 3 × 2 cm firm mass attached to the meningeal membranes of the inner left frontal bone, compressing the adjacent cerebral parenchyma [16]. In the case of the tiger in this report, it unlikely the is meningioma was the cause of the seizures due to (i) the sinonasal location of the tumour (although it did cause mild compression of the calvarium) and (ii) the fact that the seizures started occurring two years prior to the detection of the tumour. 

Most meningiomas grow as well-demarcated, firm granular masses with a broad base or pedunculated attachment to the overlying meninges, which is consistent with the afore-mentioned Bengal tiger case [16] and this case (only the attachment was to the nasal turbinates). The current WHO histological classification for meningiomas of domestic animals classifies them into two groups: slow-growing generally benign neoplasms (of eight different subtypes) and more anaplastic tumours [21], with further subtypes later identified in dogs, similar to that recognised in human meningioma [22]. The subtype in the Bengal tiger and cougar case was psammomatous (indicated by the presence of psammoma bodies and mineralization), whereas the subtype in this case was transitional. In humans, a grading system is applied to memingiomas (benign, grade I; atypical, grade II and anaplastic (malignant), grade III) [22], however, this is not the case in domestic animals. Due to the increasingly recognized limitations of the classification scheme in domestic animals, there have been attempts to establish an improved classification based on the striking similarities between canine and human meningiomas [reviewed in 9], although it is worth nothing that anaplastic meningiomas (grade III) were not detected in a study of 38 feline meningiomas [reviewed in 9], which confirms the more compressive, rather than infiltrative, nature of meningioma in cats and suggests that no currently available grading system is applicable to feline meningioma.

The most common treatment modality for cats with meningioma is surgery [8], which has been shown to significantly extend survival time after diagnosis relative to cats treated with any other strategy (with a median survival time of 685 days versus 18 days for untreated cats) [8]. However, it is worth noting that post-operative recurrence of meningioma in cats has been reported to be as much as 20% in one study, with regrowth at the site of the tumour occurring in 6/7 cases [8]. When surgical intervention and/or radiotherapy are not feasible (such as due to tumour location, patient age, etc.), medical intervention is still needed to relieve the clinical symptoms, provide quality of life and to prolong their survival time. The most common strategies to achieve this involve a combination of corticosteroids, antiepileptic medications (to reduce the frequency/severity of the seizures) and/or different chemotherapeutic agents (such as hydroxyurea) [reviewed in 9]. This was the case for the Siberian tiger in this report; treated with corticosteroids and antiepileptic drugs with a fairly good response for the first two years, until the tiger became refractory to all medication and the seizures became uncontrollable.

## 4. Conclusions

In conclusion, this is the first report of an extracranial (sinonasal) meningioma in a cat, specifically a captive wild felid. Furthermore, it highlights the similarity to that seen in meningiomas of some domestic cats; a non-infiltrative tumour with a clinically silent phenotype. It is hoped that further investigations of captive wild felids in wildlife reserves or zoological facilities will add to a better knowledge and understanding of the tumour types developed by these species, paving the way for earlier detection and successful management.

## Figures and Tables

**Figure 1 vetsci-09-00457-f001:**
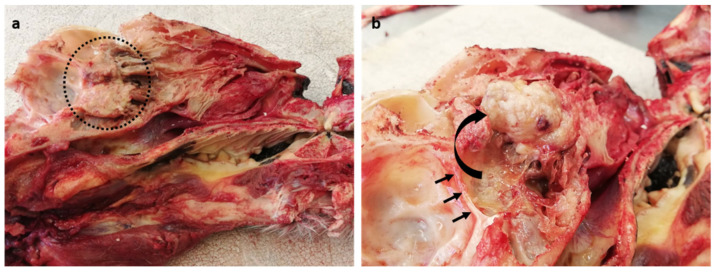
Macroscopic images of the sinonasal meningioma. (**a**) Midline section through skull and nasal cavity revealing an expansile pale neoplastic mass in the caudal nasal cavity (dashed circle). (**b**) Partial dissection and reflection of the mass rostrally (curved arrow) displaying mild compression of the calvarium (three small arrows) by the tumour.

**Figure 2 vetsci-09-00457-f002:**
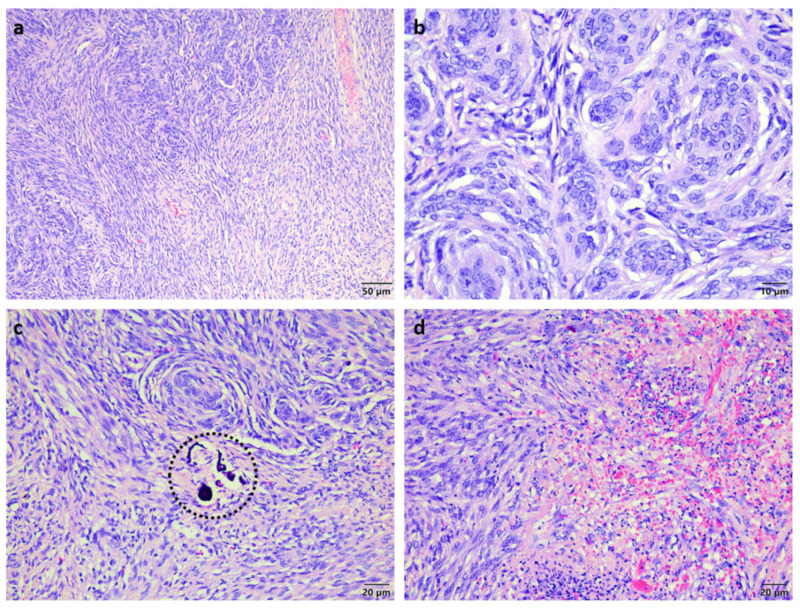
Histopathology of the sinonasal meningioma. (**a**) Low magnification revealing the transitional pattern made up of streams and whorls of neoplastic cells. (**b**) High magnification revealing the cellular morphology. (**c**) Presence of rare psammoma bodies (dashed circle). (**d**) Areas of necrosis with haemorrhage and neutrophil infiltration.

## Data Availability

Not applicable.

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
