# Peer review of "Sinonasal Meningioma in a Siberian Tiger (Panthera tigris altaica)"

_vetsci, 2022, doi:10.3390/vetsci9090457_

Round 1

Reviewer 1 Report

Descriptions of tumours of the nervous system in animals, particularly wildlife, are rare. Hence, every report is important. In the paranasal sinuses of domestic cats, lymphoma is most common, adenocarcinoma, squamous cell carcinoma and fibrosarcoma are less common.

[42] for the record, please list the brain lyres and not just mention them

[97] Instead of the name of the Idexx kits, please give specific paramaters, as not everyone uses this company's equipment and does not need to know what the Chem 17 kit contains.

[98] ProszÄ™ podać które urzÄ…dzenie Idexx byÅ‚o użyte (Catalyst One?)

Author Response

REVIEWER 1

Descriptions of tumours of the nervous system in animals, particularly wildlife, are rare. Hence, every report is important. In the paranasal sinuses of domestic cats, lymphoma is most common, adenocarcinoma, squamous cell carcinoma and fibrosarcoma are less common.

We thank the reviewer for their appreciation of our manuscript and are delighted they believe that this “report is important”.

[42] for the record, please list the brain lyres and not just mention them

As requested, we have listed the three meningeal layers. The sentence now reads: “Meningiomas are tumours that arise from the meninges, which are composed of three layers (dura mater, arachnoid mater and pia mater), and are the most common tumour of the central nervous system in humans”.

[97] Instead of the name of the Idexx kits, please give specific paramaters, as not everyone uses this company's equipment and does not need to know what the Chem 17 kit contains.

As requested the sentence has been changed and now reads: “The tiger was immobilised for evaluation by a veterinarian on three separate occasions, however, on each occasion no abnormalities were detected in the chest x-rays, pupillary light reflex tests. Blood was also taken on each occasion and analysed using a Catalyst One Chemistry Analyzer and ProCyte Dx™ (IDEXX Laboratories) for clinical chemistry profiles (including ALB, ALN/GLOB, ALKP, ALT, AMYL, BUN, BUN/CREA, Ca, CHOL, CREA, GGT, GLOB, GLU, LIPA, PHOS, TBIL and TP), electrolyte profiles (including Cl, K, Na and Na), haematology profiles (full blood count) and levels of SDMA (as a marker of kidney function), BNP (as a marker of heart failure), fasting serum bile acids, serum Amyloid A (as a marker of inflammation), total T4, total TSH, vitamin A and serum phenobarbitone; no abnormalities were detected other than elevated creatinine levels on the third occasion.”

[98] ProszÄ™ podać które urzÄ…dzenie Idexx byÅ‚o użyte (Catalyst One?)

We have used ‘Google Translate’ to translate this Polish text to English: “Please enter which Idexx device was used”.  We have now included this information (please see paragraph above).

Author Response

REVIEWER 2

This is an interesting case report that is worth publishing.

We thank the reviewer for their appreciation of our manuscript and are delighted that they believe it is “worth publishing”.

Considering the unusual presentation and location of the meningioma, various immunohistochemistry markers should have been done to further rule out carcinoma and sarcoma and confirm meningioma. If this is not possible please add the reasons why this was not done and discuss this as a limitation. (at least one study found e-cadherin a very useful marker for meningioma in cats).

The morphology of the tumour is relatively specific, especially the presence of psammoma bodies that restrict the diagnosis to meningioma, with a few very rare alternatives in humans and animals, including papillary thyroid carcinoma and papillary serous cystadenocarcinoma of the ovary. However, neither of these correspond to the location of the tumour in this tiger. We agree that performing IHC for E-cadherin would have been interesting, however, given we are the only immunohistochemistry veterinary laboratory in South Africa, it restricts us to what we have available and sadly E-cadherin is not available. Thus, based on the histomorphology, in conjunction with the post mortem findings, we feel confident with our diagnosis. 

There are also few sentences that need to be corrected or rephrased and there are some spelling errors and punctuation that need to be addressed.

Please see some specific comments below:

Line 2 the title: Please change paranasal with sinonasal. It is not clear that the mass arises form paranasal sinus or caudal nasal cavity, so including sinonasal would be better. Be consistent and change throughout the text.

As requested, the title has been changed from ‘paranasal’ to ‘sinonasal’ as well as the relevant text throughout.

Please change line 18 and 28 that there are not post-mortem confirmed meningioma arising from the paranasal cavity in cats...

As requested, these sentences have been changed to include the words “post-mortem confirmed”.

Line 71-74. Please remove this sentence it is not of relevance for the description of the case “living in a very natural environment with a pool and a forest, but had the enclosure to herself. One year later the tiger developed a mass at the dorsal right-hand side of the tail base, which was surgically removed and the histopathological examination revealed a haemangiosarcoma”.

As requested, these sentences have been removed.

Line 81. Please remove this sentence “as the owners did not want her remaining eye to be closed or removed”

As requested, this sentence has been removed.

Line 84. “Anti-histamine therapy to prevent continual corneal ulcerations”. I am not aware that anti-histamine prevent corneal ulceration..If there is evidence for this treatment add a reference or you could change due to or in case of suspected allergic etiology

Anti-inflammatory and anti-histamine treatments have been used for inflammation and allergies causing damage to eyes specifically causing deep corneal ulcers. These ulcers have found to be related to many allergic and inflammatory conditions in domestic cats and dogs and have been extrapolated for use in tigers. This is based on personal communication by Dr. Peter Caldwell with Dr. Keri-Lee Dobi (registered ophthalmologist specialist in South Africa and the UK).

Line 88 “full-blown seizures” this is not a medical word.. generalized tonic clonic seizure also include convulsion and not need to specify. Line 89-90 “fright, such as thunder, passing quad bikes or on one occasion when a gust of wind blew and a twig fell on her back.” Delete and add or freighting events.

As requested, this sentence has been modified and now reads: “In addition, generalized tonic clonic seizures also occurred; sometimes triggered by loud noises or frightening events”.

Line 92” instabilities such as twitches or tremors” other neurological signs like facia twitching and tremors. Line 93-95 “observations were carried out by the animal keepers and volunteers in person, but to a larger extent via camera traps (triggered by movement) especially when the tiger was kept in the management area for closer observation)” these sentence is too long need to be shorten and more concise.

As requested, these sentences have been modified and now read: “Observations carried out over the next two years, both by staff and movement-triggered camera traps, revealed a total of 11 seizures with convulsions, sometimes lasting up to 2 minutes, in addition to other neurological signs such as facia twitching or tremors”.

Line 96-99. Please change to blood work including hematology, biochemistry, electrolytes.. etc etc.

As reviewer 1 requested more detail on the tests that were performed, we have now modified this section as follows: “Blood was also taken on each occasion and analysed using a Catalyst One Chemistry Analyzer and ProCyte Dx™ (IDEXX Laboratories) for clinical chemistry profiles (including ALB, ALN/GLOB, ALKP, ALT, AMYL, BUN, BUN/CREA, Ca, CHOL, CREA, GGT, GLOB, GLU, LIPA, PHOS, TBIL and TP), electrolyte profiles (including Cl, K, Na and Na), haematology profiles (full blood count) and levels of SDMA (as a marker of kidney function), BNP (as a marker of heart failure), fasting serum bile acids, serum Amyloid A (as a marker of inflammation), total T4, total TSH, vitamin A and serum phenobarbitone; no abnormalities were detected other than elevated creatinine levels on the third occasion.”

Line 99 you measured phenobarbital level without giving any phenobarbital? If you start phenobarbital level pleas add the treatment with dosage on any drug the patient received before the blood test.

We apologise for the confusion. Phenobarbitone was indeed used (3mg/kg) and serum phenobarbital levels were tested for months after medication was used. To make this clearer, we have moved the paragraph relating to treatment to come before the section detailing the bloodwork.

Line 101-102 This sentence need rewriting “A brain tumour was suspected and as such the tiger was treated with oral corticosteroids antiepileptic drugs which seemed to help with varying levels of success over time as the dosages were adjusted”.

This sentence has now been rephrased and reads: “Due to a brain tumour being suspected, the tiger was treated palliatively and symptomatically with oral corticosteroids and antiepileptic drugs to reduce the rate of growth of the tumour and to keep the siezures under control. This treatment protocol seemed to help with varying levels of success with the symptoms over time as the dosages were adjusted according to the tiger’s symptoms.”

Line 104 -113“following medication was administered” medications were administered “Neurontin (Gabapentin), Medrol 104 (methylprednisolone), Lyrica (Pregabalin), Purata (Oxazepam), Phenobarbitone, 105 Clavumox (Amoxicillin), Prednisolone, Omeprazole, Keppra (Levetiracetam), Karsivan 106 (Propentofylline), SAMe Solal (s-adenosyl-l-methionine disulfate p-toluensulfonate) and 107 Liver Complex (Ingredients per capsule: Vitamin B1 (Thiamine), Vitamin B2 (Riboflavin), 108 Vitamin B3 (Nicotinamide), Vitamin B5 (Pantothenic Acid), Vitamin B6 (Pyridoxine), Vit-109 amin B12 (Cyanocobalamin), Silybum Marianum (Milk Thistle) Herb Powder, Lecithin 110 (Phospholipids), Silybum Marianum (Milk Thistle) Seed Extract, Cynara Scolymus (Arti-111 choke) Leaf Powder Extract 4:1, Schisandra spp. Fruit Powder, Hordeum Vulgare (Barley) Leaf Powder (SOD Precursor), Taraxacum Officinale (Dandelion) Root Powder, L-Gluta-113 thione, Selenium) and Gingko biloba”.

Please follow the journal standard for the medications, add the dose for each medication and the interval, remove all the supplements and add various supplement were administered as well.

As requested, we have provided the details for the medications used and the sentence now reads: “Neurontin (600mg/tab Gabapentin, Pfizer; starting on 5mg/kg twice a day (bid) for 5 days, then 10mg/kg bid, then tapering to once a day (oid) after two weeks, maintained at this dose for 4 months), Medrol 104 (16mg/tablet methylprednisolone; 1mg/kg bid for 5 days, then 1mg/kg oid for 14 days, 0.5mg/kg oid for 14 days, 0.5mg/kg every other day (eod) for 1month), Lyrica (75mg/tab Pregabalin; 1mg/kg bid for 3 weeks), Purata (30mg/tab Oxazepam, Aspen; 0.2mg/kg bid for 10 days, then 0.2mg/kg oid for 10 days, then 0.2mg/kg three times a week), Phenobarbitone (30mg/tablet BeTabs Sedabarb; 3mg/kg for 1 year), Clavumox (Amoxicillin/clavulanic acid 1g/tablet; 12.5mg/kg bid for 14 days), Prednisolone (5mg/tablet, Aspen; 1mg/kg bid for 5 days, then 1mg/kg oid for 14 days, 0.5mg/kg oid for 14 days, 0.5mg/kg eod for 1 month), Omeprazole 40mg/tablet (compounded vetscripts tablets; 0.5mg/kg eod for 6 months), Keppra (750mg/tablet Levetiracetam, Glaxo Smith Kline; 20mg/kg bid for one year), Karsivan 106 (50mg Propentofylline/tablet, MSD animal health; 3 tablets bid) and various supplements to support the liver, including SAMe Solal (s-adenosyl-l-methionine disulfate p-toluensulfonate) and Liver Complex (Ingredients per capsule: Vitamin B1 (Thiamine), Vitamin B2 (Riboflavin), Vitamin B3 (Nicotinamide), Vitamin B5 (Pantothenic Acid), Vitamin B6 (Pyridoxine), Vitamin B12 (Cyanocobalamin), Silybum Marianum (Milk Thistle) Herb Powder, Lecithin (Phospholipids), Silybum Marianum (Milk Thistle) Seed Extract, Cynara Scolymus (Artichoke) Leaf Powder Extract 4:1, Schisandra spp. Fruit Powder, Hordeum Vulgare (Barley) Leaf Powder (SOD Precursor), Taraxacum Officinale (Dandelion) Root Powder, L-Glutathione, Selenium) and Gingko biloba.

Line 115-117 “Thus after several days of inappetence, lethargy, difficulty in administering medication and unresponsiveness to medication, the decision was taken to euthanize the tiger due to deteriorating condition” this sentence need rephrasing.

As requested, this sentence has been modified and now reads: “Thus after several days of inappetence, lethargy, difficulty in administering medication and unresponsiveness to the medication, the decision was taken to euthanise the tiger.”

Line 138 “mas rostrally” change to Mass rostrally

This typographical error in Figure 1 legend has now been fixed.

Line 178 The fact that a tumor do not metastases do not make it a benign one, meningioma are not benign tumors as they kill the patient if not treated appropriately. “benign nature of meningiomas in dogs and cats” Please changes this sentence. Line 181, same as above “slow growing and thus considered ‘benign’, with no invasion into the underlying brain parenchyma” .You can change to “slow growing with rare infiltrative growth and invasion into the underlying brain parenchyma”. Remove benign form the text use other words to describe the low grade, expansive rather than infiltrative growth nature of most meningioma.

As requested, these sentences have been modified and now read: “Meningioma is the most common primary brain tumour in domestic cats and is typically slow growing slow growing with rare infiltrative growth and invasion into the underlying brain parenchyma. Nevertheless, although infiltrative meningiomas are rarely seen in cats, they can co-occur with non-infiltrative meningiomas”. However, we have kept the word “benign” when referring to the WHO classification system of meningiomas in domestic animals and humans grading system as that is the terminology that is currently used.

Line 206-209 “In the case of the tiger in this report, it is unclear whether the meningioma was the cause of the seizures due to its location in the paranasal cavity, since the tumour did cause mild compression of the calvarium; however, the seizures started already two years earlier (at which stage the meningioma may not have been present)” please rephrase.

As requested, this sentence has been modified and now reads: “In the case of the tiger in this report, it unlikely the is meningioma was the cause of the seizures due to (i) the sinonasal location of the tumour (although it did cause mild compression of the calvarium) and (ii) the fact that the seizures started occurring two years prior to the detection of the tumour”.

Line 234-241 please rephrase: “When surgical intervention and/or radiotherapy are not feasible (such as due to tumour location, patient age, etc), medical intervention is still needed to relieve the clinical symptoms, provide quality of life and to prolong their survival time, and the most common strategies involve a combination of corticosteroids, antiepileptic medications (to reduce the frequency/severity of the seizures) and/or different chemotherapeutic agents (such as hydroxyurea) [reviewed in 9], as was the case for the Siberian tiger in this case report (treated with corticosteroids and antiepileptic drugs with a fairly good response for the first two years, until the tiger became refractory to all medication and the seizures became uncontrollable)”

As requested, this sentence has been rephrased and broken into three sentences: “When surgical intervention and/or radiotherapy are not feasible (such as due to tumour location, patient age, etc), medical intervention is still needed to relieve the clinical symptoms, provide quality of life and to prolong their survival time. The most common strategies to achieve this involve a combination of corticosteroids, antiepileptic medications (to reduce the frequency/severity of the seizures) and/or different chemotherapeutic agents (such as hydroxyurea) [reviewed in 9]. This was the case for the Siberian tiger in this report; treated with corticosteroids and antiepileptic drugs with a fairly good response for the first two years, until the tiger became refractory to all medication and the seizures became uncontrollable.”

References: Please review the reference style, at least 3 references are not in the format of the journal standard

The references have all been reviewed and the ones not in the correct format have now been modified.

Round 2

Reviewer 1 Report

After corrections the paper can be published